# A scoping review of the residual barriers to skilled birth attendance in Ghana: A conceptual framework and a fish bone analysis

Juliet Abredu [1,2]*, Catherine K. Dwumfour [3], Boo Alipitio [3], Mawusi Alordey[1], Veronica Millicent Dzomeku[3], Sophie Witter [2]

**1** Ho Nurses' Training College, Ho, Ghana, **2** Institute for Global Health and Development, Queen Margaret University, Edinburgh, United Kingdom, **3** Department of Nursing, College of Health Sciences, Kwame Nkrumah University of Science and Technology, Kumasi, Ghana

* julietabredu@yahoo.com

## Abstract

The achievement of the Sustainable Development Goals (SDGs) targets 3.1, 3.2 and 3.3.1 is strongly dependent on the effective utilization of skilled birth services. Despite advancements made in Skilled Birth Attendance (SBA) in Ghana, there are still instances of unassisted childbirths taking place. The aim of this study was to explore the residual barriers of SBA such as community- and health system-related factors affecting SBA in Ghana and to identify strategies for addressing them. An electronic search was done using PubMed, Popline, Science direct, BioMed Central, Scopus and Google scholar for peer reviewed articles as well as grey articles from other relevant sources, published between 200 and 2022 on community- and health system related factors influencing SBA in Ghana. Out of the 89 articles retrieved for full screening, a total of 52 peer-reviewed articles and 1 grey article were selected for the final review. The study revealed that cultural practices (community factors), low quality of service delivery due to the inappropriate behaviors, lack of competency of skilled birth attendants (SBAs) as well as the inefficient distribution of SBAs contribute to ineffective uptake of SBA (health system factors). Also, indirect costs are associated with the utilization of skilled delivery care even with the existence of 'free' delivery care policy under the national health insurance (policy factor). For Ghana to achieve the SDGs above and improve SBA, it is essential to enhance the quality of skilled delivery care by addressing the attitude and competencies of skilled birth professionals, while plans are put in place to expand and develop the Community-based Health Planning and Services (CHPS) strategy to help address the access barriers to SBA. More so, the 'free' delivery care policy should absorb all the costs associated with skilled delivery for pregnant women as it is intended for.

**Data Availability Statement:** All relevant data are within the paper and its Supporting Information files.

**Funding:** The authors received no specific funding for this work.

**Competing interests:** The authors have declared that no competing interests exist.

## Introduction

Globally, approximately 810 women die every day from causes related to pregnancy and child-birth, with 94% of the maternal and child deaths occurring in developing countries, including Ghana [1]. Skilled attendance during delivery is considered the most crucial factor in preventing maternal deaths [2] and neonatal deaths in less developed countries [3]. Globally, the target for Skilled Birth Attendance (SBA) is set at 90% but Ghana set a target of 80% [4] to make it more realistic to achieve.

Good progress has been attained in SBA in Ghana, from 44 percent in 1993 to 68 percent in 2011, and with the current SBA of 79 percent. This is after the implementation of the free maternal and delivery services and the Community-based Health Planning Services (CHPS) [1]. Despite this progress, unsupervised deliveries still occur. The findings of the 2017 Ghana Maternal Health Survey indicated that the vast majority (98%) of maternal deaths occurred in women who had attended less than four antenatal care (ANC) visits. Additionally, only 37.4% of the women who died had received skilled assistance during delivery [5]. About 13–33 percent of maternal deaths and two thirds of neonatal deaths could have been averted when births are attended by skilled birth professionals [3]. This is why it is important that all expectant mothers have access to skilled delivery [2].

Some of the commonly cited factors which hinder women from having skilled delivery are distance to a health facility, precipitated labour, poor-quality care, and sociocultural factors like getting consent from others before having skilled delivery [6,7]. This calls for much more focused interventions in dealing with the residual barriers to skilled delivery by women in Ghana and ensuring that, the Sustainable Development Goals (SDGs) targets 3.1, 3.2 and 3.3.1. are achieved. A proposed conceptual framework was developed in this study for analyzing these factors in the Ghanaian context. For example, a framework developed in Ghana by Aful-ani and Moyer [8], analysed the frameworks by Thaddeus and Maine [9] and Gabrysch and Campbell [2] on the 'three delays' in seeking care by pregnant women, by using it to explore how socioeconomic factors can affect the delays in receiving adequate care. Similarly, the framework by Esena and Sappor [10] also adapted Gabrysch's model to analyse factors influencing facility delivery in Ghana. The limitation of all these frameworks mentioned above is that, they considered supply and demand side factors and how they impact on seeking skilled delivery care, but they did not incorporate delivery care policy related factors and its impact on SBA. In this scoping review, we explored the residual community and health system barriers along with its related policy barriers to SBA in Ghana.

## Methodology

### Study design

This study was a scoping review of literature on residual barriers to SBA in the Ghanaian context, and it was based on the PRISMA Extension for Scoping Reviews (PRISMA-ScR) guidelines [11]. This section describes the methodology used in the selection of articles in answering the research question and explains the analytical framework that was used.

### Research objectives

1. To identify the health system and its related policy factors that influence uptake of SBA.

2. To explore the community factors influencing SBA.

3. To propose effective interventions to accelerate efforts towards achieving targets for SBA in Ghana.

## Literature search

The following data bases and webpages were utilised to conduct the electronic search: National Health Insurance Scheme (NHIS), World Health Organization (WHO) and other websites; PubMed, Science direct, Scopus, Popline, BioMed Central, google scholar. In addition, manual search was conducted to identify relevant references from the included studies and the full text of articles retrieved. Furthermore, the guidance of scholars in the field was sought, for the right source of scholarly materials to be included.

The search was conducted using the following keywords: skilled birth attendant/attendance, skilled birth utilization, facility delivery, home delivery, maternal mortality, National Health Insurance Scheme, Free delivery care policy, and Ghana. These terms were used in combination of terms such as 'predictors' or 'determinants' or 'factors influencing' and 'Ghana'.

## Eligibility criteria

**Inclusion criteria.** Included in the search were peer reviewed publications and grey literature written in English Language and were published between the years 2005 and 2022. The papers were selected from 2005 due to the nationwide implementation of Ghana's Free Maternal Health Care Policy (FMHCP) and its integration into the National Health Insurance Scheme in 2008. Additionally, articles up to 2022 were included to ensure the incorporation of current information on the topic. The study included articles specifically focused on Ghana, encompassing both descriptive and evaluative studies.

**Exclusion criteria.** Articles that did not meet the inclusion criteria such as those written in languages other than English or with publication dates prior to 2005 were excluded from the study.

## Study selection

The databases were searched by the research team from 1st January 2005 to 31 December 2022, for peer reviewed articles related to the topic. Relevant websites were also searched for grey literature and technical reports. Irrelevant and duplicate articles were removed from the data set to ensure that only relevant articles were included for further analysis. The remaining articles were evaluated for eligibility by reviewing their abstracts. Additionally, manual searching of the reference list of the included articles was done to identify additional relevant articles. Afterwards, articles that were deemed irrelevant for the study were excluded from the pool of fully scrutinized and hand-searched articles. Finally, peer reviewed articles and grey literature, including technical reports that met the specific inclusion criteria were used for the study (Fig 1).

## Quality appraisal of study methods

The authors evaluated the quality of the reviewed articles by utilizing a modified critical appraisal checklist that was originally developed by the Joanna Briggs Institute [12], as described in [13].

## Charting the data

The data extracted from the study included bibliographic details, study setting and population, study design, study objectives, type of barrier identified, outcome measures and relevant results. The synthesised data allowed the research team to derive themes that were instrumental in shaping and ensuing discussion on the findings (Table 1). The analysis of data was done in the narrative form.

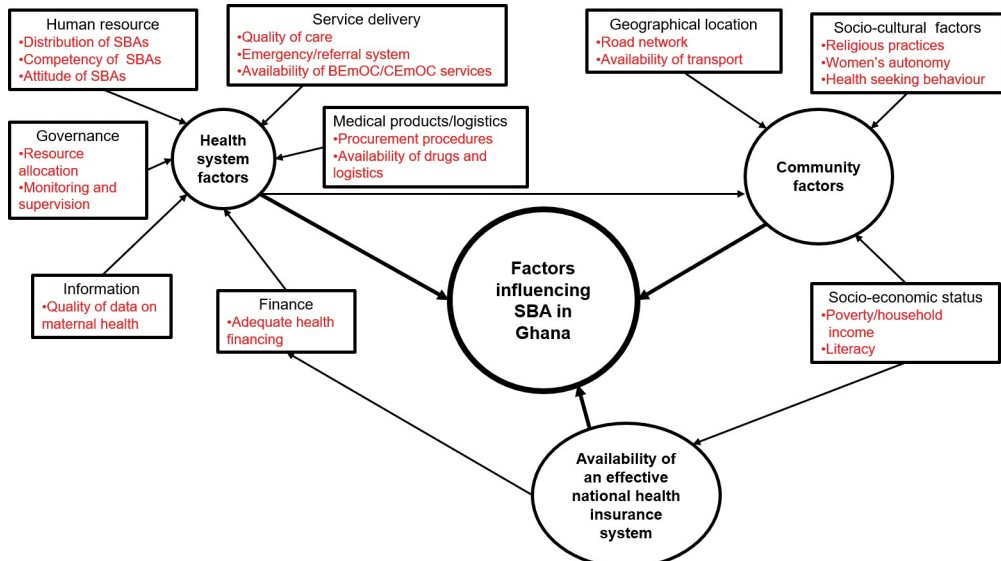

**Fig 1. PRISMA-SCR Flow diagram of literature selection.** Grey and peer reviewed articles date between 2005 and 2022.

## A framework of factors influencing SBA

To guide the organisation and analysis of the articles identified, a conceptual framework of factors influencing SBA in the Ghanaian context was developed by the research team. This framework (Fig 2) builds on the earlier frameworks [8,10] on SBA in Ghana, and applied the health system building blocks' framework by WHO [62] to the Ghanaian context. Additionally, supply and demand side and policy factors (national health insurance) were integrated based on preliminary knowledge on the topic. Thus, two main themes were identified: health systems and its related policy factors as well as, community factors influencing SBA in Ghana (Fig 2).

**Table 1. The themes and sub-themes identified from reviewed studies.**

| Themes | Subthemes | References |
|---|---|---|
| Health systems factors | Health service delivery | [7,14–22] |
| | Health workforce | [7,14,15,17–19,21–36] |
| | Health information system | [18,37] |
| | Medical products, equipment, and infrastructure | [18,19,24,26,29,38–41] |
| | Health financing | [42,43] |
| Policy factor under health Systems | NHIS sustainabilty | [17,40–44] |
| | Governance/Leadership | [14,24,25,45] |
| Community factors | Geographical barriers to SBA | [15–17,32,35,46–52] |
| | Sociocultural barriers to SBA | [7,14,26,31,35,53] |
| | Religious and cultural practices | [7,14,26,31,35,53] |
| | Preference for Traditional Birth Attendants | [15,17,33,35,54,55] |
| | Women's decision-making autonomy | [52,53,56–58] |
| | Socioeconomic barriers to SBA | [6–8,10,32,48,49,53,55,59–61] |

NHIS; National Health Insurance Scheme, SBA; Skilled Birth Attendance.

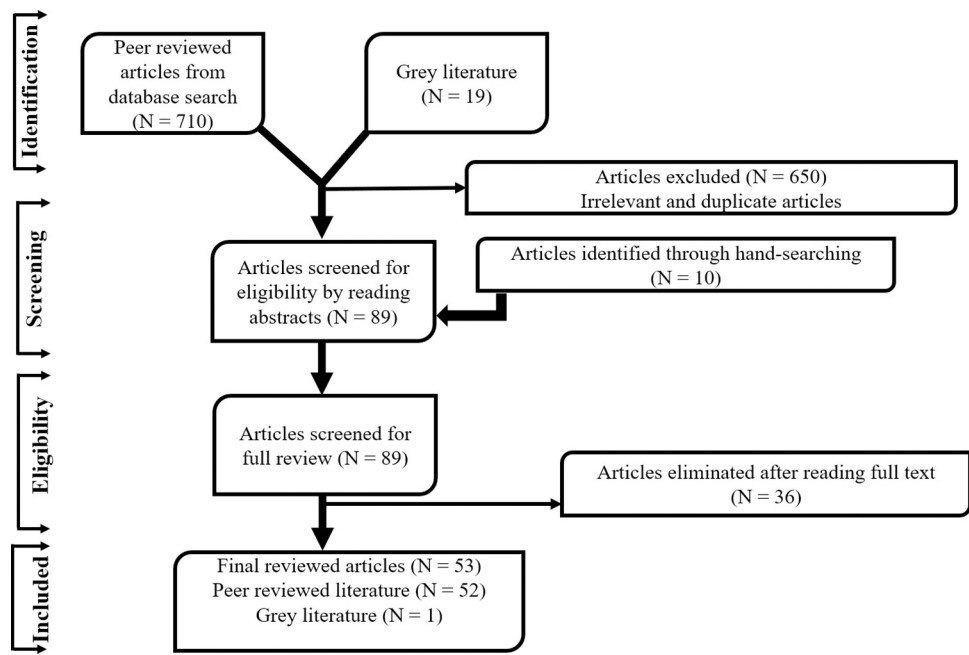

**Fig 2. A conceptual Framework of factors influencing SBA in Ghana.** The framework shows the interactions of these factors and how they influence SBA in the Ghanaian context.

## Results

### Characteristics of included studies

Out of the 53 articles used for the final review, 52 were peer reviewed articles and 1 was grey literature. Out of the peer reviewed articles, 49 were solely conducted in Ghana while 3 focused on Ghana along with other African countries. Among the peer reviewed articles, 23 explicitly conducted primary data analysis (quantitative (5), qualitative (15), mixed methods (3)). Additionally, 17 used explicitly secondary data from the Ghana Demographic Health Survey (GDHS), regional and district Health level data; 2 used a combination of secondary data together with qualitative methods; 2 were explicitly review studies; 1 combined a review with qualitative method. The remaining peer reviewed articles consisted of policy note (1); book (1); technical report (3) and commentary (2). The grey literature was a report from a national newspaper.

### Health systems factors

The availabity of health services to meet the required quality and standard is an essential function of a health system. Health service delivery is therefore an immediate output of all other health system inputs such as medical products, health workforce and finance [62].

### Health service delivery

In Ghana, lack of proximity to health facilities in some parts of the country is a hindrance to utilization of skilled delivery [14–16]. In the Upper West Region (UW/R) for example, the estimated average distance from the house to a health facility is more than 8 km for three quarters of households [17]. Similarly, in the Upper East Region (UE/R), the average distance from the community centres to a health facility is estimated at 18 km [14]. Also, Nesbitt, Lohela [16]

used secondary data to assess the effects of the quality of health service delivery and distance on facilty delivery in seven rural districts in the Brong Ahafo Region between 2008–2009 and found that 58% of the deliveries were facility deliveries out of which 79% of women who had the facility delivery lived within 1km distance to health facilities, while 28% lived more than 10km revealing the effect of distance on facility delivery.

Also, a cross-sectional study was conducted by Nakua, Sevugu [15], in Amansie West district of the Ashanti Region (A/R) which is predominantly a rural district with only one comprehensive emergency obstetric care (CEmOC) facilty serving about 162 communities in 2012. The aim of the study was to determine the barriers to SBA in the rural district, despite a Milenium Village Project (MVP) initiated in the district in 2006 to improve uptake of skilled delivery following a 29% SBA recorded in the district. The study found an improvement in SBA in the MVP intervention communities, however even though most of the women in the study area were reported to be living close to health facilities of less than 5km, almost 50% of them still had home deliveries and some of the reasons stated were poor attitude from health workers, lack of education on the importance of skilled delivery, availabilty of Traditional Birth Attendants (TBAs), and transportation challenges. Some studies mentioned the lack of a well coordinated and functional national referral system coupled with inadequate and expensive ambulance services in the country, as a health service delivery challenge that deprives some expectant women from accessing basic emergency obstetric care (BEmOC) or CEmOC services during emergency situations [18–20].

Avortri and Modiba [21] also conducted a qualitative study involving 15 women from two district hospitals in Greater Accra Region (GA/R) to explore factors influencing the quality of skilled delivery services and ways to improve these services in the country. The study revealed some health service delivery challenges such as poor organization of services, unavailabilty of some services, poor attitude of staff, pressure of workload on staff, coupled with shortages of SBAs as factors that hinder the quality of skilled delivery care provision. They however suggested that the health system should organize service delivery to promote client centred and friendly relationship between providers and their clients.

Further, Rominski, Lori [22] conducted a qualitative study using focus group discussion involving 15 final year midwifery students in the 10 regions of the country between September 2013 and Febuary 2013. The aim of the study was to understand reasons for abusive behaviour of SBAs on women during delivery. The students acknowledeged the importance of quality care but also justified the reasons for abusive treatment by SBAs on labouring women to be mainly due to lack of cooperation from the women during labor which could result in detrimental birth outcomes. Similarly, Moyer, Adongo [7] also conducted a qualitative study in the the Kassena-Nankana District in the Northern Region (N/R), between July and October 2010 with 128 participants (community members, TBAs, health professionals, compound heads,traditional healers) using in-depth interviews as well as focus group discussions to explore the attitudes and perception of community members and health care providers towards abusive behaviour during delivery in this setting. Although most of the study participants commented on positive experiences during skilled delivery, almost all respondents, including health care providers, admitted that maltreatment, discrimation, physical and verbal abuse of women are major problems labouring women experience in this setting.

**Health workforce.** An SBA is any professional for example, a nurse, doctor or midwife, competent and proficient in midwifery skill, to manage and conduct normal deliveries as well as refer complications [23]. Many studies conducted in Ghana, highlight the shortage of SBAs as an eminent problem of the health system [15,18,19,22,25–27]. The shortage is in part due to the emigration of doctors and nurses, largely to the United Kingdom and the United States [24]. In connection with this, the government has increased the training of SBAs, consequently

leading to an increase in the percentage of nurses and midwives from 58% to 370% between 2008 and 2018. Also, the density of SBAs thus, doctors, nurses and midwives has improved significantly from 1.07 per 1000 population to 2.5 per 1000 population between 2005 and 2017 [25].

Notwithstanding, the distribution of SBAs remain a major challenge [24,25]. In Ghana, the distribution of health workforce including SBAs, is skewed towards the southern parts of the country which are considered the most affluent regions compared to the northern parts [25,28]. There are also disparities in the rural/urban distribution of health workers in the country. According to GHWO [27], about 70% of high cadre health workers example medical doctors, professional nurses/midwives, and pharmacists are situated in the urban areas which implies that, there are only few highly skilled professionals in the rural areas to provide health service delivery. More so, Appiah-Denkyira, Herbst [24] mentioned that, the prolong delay by the government in recruiting SBAs to fill vacant positions also contributes to shortages of these SBAs especially in areas where their services would have been needed immediately. However, some highly skilled SBAs also refuse postings to rural areas which also explain the reasons for the shortages of these cadres of skilled professionals in remote areas. Some reasons stated for refusal of posting to these areas are lack of career-based incentives and professional development opportunities, poor health infrastructure and working conditions, as well as delay or refusal of granting transfers to urban areas after working in the rural area for a long period.

Additionally, lack of competence of some SBAs in the management of obstetric cases has been widely reported as one of the health systems challenges that should be critically looked into [15,19,25,29]. In the study of Lohela, Nesbitt [29], they revealed that, lack of competent SBAs might hinder service delivery even more than lack of medical products and logistics. Appiah-Denkyira, Herbst [24] and Asamani, Amertil [25] concur with the above, further stating that, the quality of training nurses and midwives is deficient in infrastructure and technical capacities, thus contributing to the lack of competencies. This is further aggravated by limited access to in-service trainings and workshops for SBAs to enhance practical proficiency in managing maternal conditions and complications [19,25,26].

Asamani, Amertil [25] documents an interesting finding in their studies, which revealed that, before 2008, about 88% of midwives were older than 46 years compared to 3% being less than 35 years, which called for the introduction of the direct midwifery training to compensate for future attrition in the workforce from retirement. Consequently, the demographic shift has changed significantly in 2018, where about 71% of midwives were between the ages of 25 to 35 years and just 12% were older than 46 years. The implication is that the older midwives are perceived by the communities to be more experienced than the younger ones in the management of obstetric cases but have almost faded from the system which might impact on quality of skilled delivery. For example in the study of Rishworth, Dixon [17] some of the participants (midwifery students) mentioned that, most laboring women do not comply with service delivery protocols because they feel they are old enough to even be mothers of the midwives attending to them. This suggests that recruitment of SBAs should be well mixed, especially in terms of age categories, considering the Ghanaian culture where values and norms are held in high esteem by the society.

Besides the problem of incompetence, unacceptable attitudes of SBAs are also widely reported barriers to skilled delivery. Some of which are: unofficial demands from pregnant women, for example soap and other disinfectants after delivery [7,15,16,19,20,22,23,31–36] and abusive behaviors [7,23,37]. Further, high workloads have also been documented as one of the challenges that SBAs experience which needs to be compensated for, by the provision of additional incentives [25,26].

**Health information system.** A quarterly verification of data was introduced for all regions to implement, as efforts to ensure and promote the quality and timely reporting of

data on health. Monthly feedback is also provided on performances at the regional levels regarding completeness and timeliness of data submitted, which are publicly made available and shared with other regions. Also, the Ghana health service in partnership with Christian Health Association of Ghana (CHAG) have initiated guidelines for data verification that will promote data quality as well as ensure standardized performances [37]. However, data collection and storage on obstetric care, especially at the peripheral levels in the health system, is a major challenge which affects planning and allocation of resources for improving maternal care. Both the public and private facilities are often unable to provide reliable data which makes it difficult to link data to service provision [18]. There is also evidence of low commitment among some senior officials at all levels of the health system in supporting the use of District Health Information System (DHIMS) data for reporting, as well as for decision making [37].

**Medical products, equipment, and infrastructure.** The procurement of medical products in Ghana is decentralized from the national central stores and controlled by the Ministry of Health (MOH) procurement department, through the regional and district stores to the facilities. And these medical products are usually procured from 'revolving drug funds' of facilities. However, there are challenges in the procurement procedures often due to shortages of drugs from the central stores, and high cost of drugs which causes most facilities to procure essential drugs from private providers which might not be of good quality [39,40]. Additionally, Bossert, Bowser [38], revealed that existing national guidelines on inventory control of medical logistics were not being adhered to in most facilities and these together with poor procurement decisions, led to poorer health performance. Lack of adequate supply of drugs on the National Health Insurance drug list in health facilities, have a detrimental effect on the quality-of-service delivery, including skilled delivery care [20,41].

Some studies also mentioned inadequate medical equipment and infrastructure such as limited space in delivery rooms, and poor facility conditions, as other eminent challenges [19,25,27,42]. In the rural areas most especially where the problem of lack of medical supplies and equipment is more profound, it undermines the provision of quality maternal health care [24]. Nonetheless, some studies also report a general improvement in the availability of required drugs and equipment [30,42].

**Health financing.** During the mid-2000s, Ghana was close to achieving the target of 15% allocation of public expenditure on health, however, this has reduced in recent times. The national health budget has increased largely as a result of increasing expenditure on salaries estimated at 94%, and less than 10% on goods and services. Since 2012 to 2017, the general government expenditure on health was 2.8%, but in addition to National Health Insurance Fund (NHIF) (5.9%) and Internally Generated Fund (IGF), it added up to an average of 9.3%. This generally reveals government's relatively low investment into the health system [42]. Public health facilities thus, rely mostly on IGF to sustain the continuous provision of health services to the people [43].

*NHIS sustainabilty.* The NHIF adds significantly to the health sector budget but is not growing at pace with the rest of the MOH budget [39]. Other sources of funding for health such as donor contributions also remain small [40]. This affects the funding of the free delivery care policy under the NHIS thus to some extent threatens its sustainability [41]. This also leads to delays in payment of claims to service providers thus, results in informal charges which subsequently impacts on the provision of quality maternal health care including skilled delivery care [17,43]. For instance, in a report, the Ghana Medical Association (GMA) called on the government and NHIA to urgently pay claims that were due for September 2018 to May 2019 or else providers will resort to 'cash and carry' system [44].

**Governance/Leadership.** The MOH performs the overall leadership role over the health system, however, authority is delegated to the Ghana Health Service (GHS) and CHAG [24].

The health service administrative organizational hierarchy begins from top to bottom thus; central, regional, district and sub-districts which also corresponds to the political administrative structure [24]. There are reported failures in management practices which impacts on the health system in general and affects the provision of quality maternal health care [15,25].

Some studies highlight the inefficient budgeting for health service as one management challenge which affects allocation of resources [15,46]. Evidence suggests that, the decision of resources allocation to the health sector is politically influenced, an example of which is seen in the government's commitment in investing more in building big hospitals while the CHPS strategies remain highly underfunded [45].

Appiah-Denkyira, Herbst [24] illustrates the ineffective decentralization of authority over human resource management in the GHS unlike CHAG, especially in recruitment of health workforce, as a challenging issue which affects effective management of human resource.

There are also concerns about poor supervision of leaders across all management levels in the health system, which is reflected in the poor quality of service provision and the poor attitude of health workers [24].

Asamani, Amertil [25] particularly noted that some facilities have upgraded junior nurses to the level of nurse managers or ward in-charges without any prior leadership preparations, thus might undermine their ability to perform leadership roles effectively. These may affect the organization of services and quality of care provided, thus subsequently impacting on SBA.

## Community factors

**Geographical barriers to SBA.**  Geographical accessibility such as distance, poor road networks and lack of transport to health facilities, have been widely reported as factors associated with home deliveries in some parts of the country [17,33,36,47–50]. Some women especially those living in the rural regions are faced with challenges accessing skilled delivery due to supply side issues such as lack of health facilities and personnel [16,18,51]. Meanwhile, Millogo, Musenge [51] found a sharp decline in home births particularly in the rural areas than the urban districts of Kassena-Nankana districts in the UE/R, between 2003 and 2009, which may partly be due to the fact that the CHPS programme was first implemented in that district. Similarly, Gudu and Addo [52] reported high uptake of skilled delivery in a rural district in UE/R.

**Sociocultural barriers to SBA.**  *Religious and cultural practices.* Further, some women are hindered by religious and cultural factors which limit their access to SBA [14]. For example, among some cultural groups, it is considered a 'taboo' to deliver in a hospital, thus, laboring women are required to deliver in a place which ensures adequate privacy such as homes and where cultural practices surrounding birth can be performed [7,15,54]. Women who violate this and go on to have facility deliveries are expected to welcome misfortunes during birth [26]. Home births are reported to give a woman some prestige in certain communities since they are considered strong and brave, thus a woman who delivers in health facilities are often regarded weak, irresponsible or perceived to be engaged in infidelity [15,27,32,54]. Additionally, some women prefer home delivery because they believe that the way the placenta is discarded following childbirth significantly impacts the prospects of the offspring in the future. This is because the placentas are usually burnt in the health facilities rather than giving the women or their families the opportunity to bury them in their homes [35]. Others also choose home deliveries because of the fear of caesarean section or assisted delivery [26].

*Preference for traditional birth attendants.* The existence of TBAs in most communities (despite the ban imposed on them from conducting deliveries), also influences people's decision to have unskilled deliveries. This is because, deliveries with TBAs are perceived to be associated with culturally appropriate birth practices and an alternate source for unavailability of

SBAs as well as distance and cost barriers to accessing skilled care. More so, some women have perception of poor quality of skilled delivery care thus, prefer home deliveries [16,18,34,36,55]. Haruna, Kansanga [55] and Rishworth, Dixon [17], therefore, strongly argue for the re-training of TBAs in the light of inadequacy of SBAs and integrating them into the health system as a sound strategy to improve facility deliveries, promote healthy cultural practices and reduce maternal mortalities.

*Women's decision-making autonomy.* Also, women are reported to lack autonomy in decisions regarding seeking skilled attendant at delivery, thus are influenced largely by their husbands and households [54,57]. However, Gudu and Addo [52] in their study found that majority of the women made the decision themselves to have skilled delivery. Similarly, Cofie, Barrington [57] and Cofie, Barrington [58] in their study found that the availability of a network of family and social support, improved women's uptake of facility delivery.

**Socioeconomic barriers to SBA.** Socio-economic characteristics such as low education, unemployment, and poverty are some cited factors for unskilled deliveries [50,60–62]. It is reported that women who are older, multiparous, do not have existing medical conditions or complications in pregnancy, practices Islamic or traditional religions or belong to minority ethnicity are also less likely to seek skilled delivery [8,60,61]. Further, many studies highlight the cost associated with skilled delivery despite the 'free' delivery care policy under the national health insurance, such as the cost involved in buying delivery items (layette), as the main determinant to accessing skilled attendant at birth considering the low household income status of families [7,10,20,33,49]. In this regards, Haruna, Kansanga [55] suggested that women should be given incentives such as a small pack containing detergents and baby accessories in health facilities to encourage them to have skilled delivery. Moyer, Adongo [53] revealed that while socioeconomic factor serves as a major determinant of skilled delivery especially in the rural north of Ghana, it is often a mediator of sociocultural factors.

Additionally, evidence available from the 2014 GDHS on the most serious barriers to SBA showed that, out of the total respondents (women) interviewed, 51% admitted they encountered at least one barrier to accessing health care including SBA. Among those respondents, 42% complained of (financial barriers), 25% (distance barriers), 16% (lack of companion to health facility) and only 6% complained about (lack of autonomy) [6].

Fig 3 is the graphical presentation of the findings from this review, in a fishbone diagram. Fishbone/Ishikawa diagram is also referred to as the cause and effect diagram and it is originally generated by Kaoru Ishikawa [63]. This fishbone diagram will provide a graphical understanding of the factors that influence SBA in Ghana.

## Discussion

This scoping review explored the health system and its policy-related factors and community factors influencing SBA in Ghana. Some health system challenges have contributed to the ineffective uptake of skilled delivery in Ghana. There is ineffective distribution of health facilities in the country. The rural areas most especially lack BEmOC facilities which have resulted in most users having to travel long distances to access skilled delivery. This is in line with a systematic review study conducted by Wong, Benova [64] which assessed the impact of geographical accessibility to skilled delivery in sub Saharan Africa and found that, longer distance to health facility negatively impacts on skilled delivery and the impact is more pronounced for women in the rural areas. In Ghana, this is even worsened by the unequal distribution of SBAs, and it is due to the poor management and recruitment of health workforce and lack of incentives attached to working in the rural areas. Even though the government has introduced the CHPS programme and increased the intake of midwives in training as a strategy to

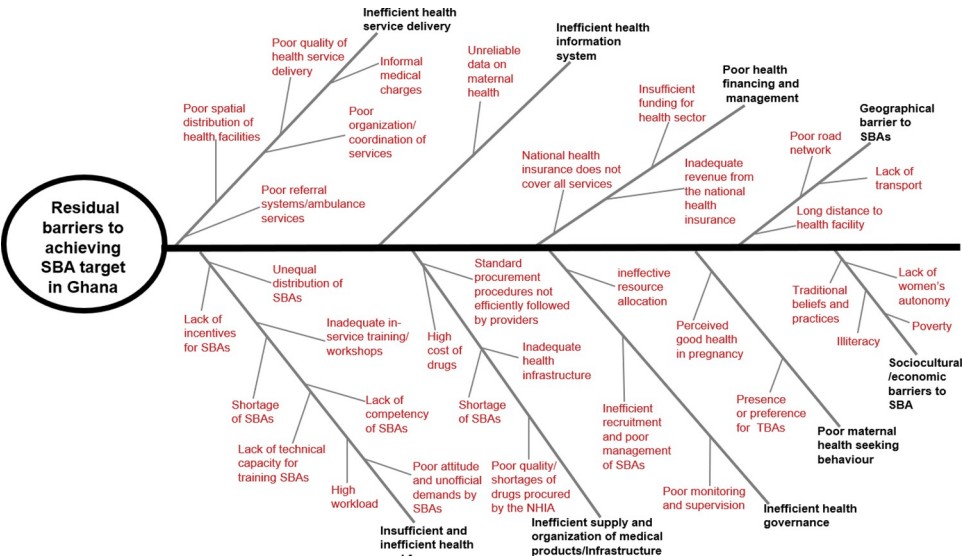

**Fig 3. Fishbone analysis of the residual barriers to achieving SBA target in Ghana.** The inscriptions in bold, have been identified as the major factors influencing the utilization of SBA in Ghana. Poor health financing and management, inefficient health governance, inefficient supply and organization of medical products/infrastructure, couple with insufficient and inefficient health workforce, results in the overall inefficient health service delivery, such as poor organization/coordination of services and poor quality of health service delivery. Other community factors such as geographical, sociocultural, and economic barriers to SBA, and poor maternal health seeking behavior also impact on SBA in Ghana.

improve SBA, the unavailability of midwives, especially in CHPS centres in the country, makes this impossible [24].

Similar finding is reported in Malawi where there is uneven distribution of professionals to provide obstetric care at both the district and health centre levels [65]. This may not be the case in all other countries though, but it is important for Ghana to look critically into these problems of unavailability of SBAs and BEmOC services especially in the remote areas, as this can also prevent some women from enrolling onto the national health insurance scheme. This is because; they may not see the need for enrolling if health facilities are not available to provide skilled delivery care, and this can negatively impact on SBA.

Additionally, the provision of skilled delivery care is shown in this review to be ineffective in the country. Some SBAs in the country lack competence and proficiency in obstetric care. Apart from that, the attitude of SBAs is considered poor and unprofessional, especially in relation to customer care practices. This is especially true where SBAs have been found to be discriminatory and abusive to their clients, especially during labour. This is consistent with the study of Ayanore, Pavlova [18] conducted in Ghana and Gebremichael, Worku [66] conducted in northern Ethiopia. In addition, the high cost and poor coordination of emergency referral services and the indirect costs associated with utilization of skilled delivery even with the 'free' delivery care policy is also a hindrance to the provision of effective skilled delivery care and is in line with the findings of Abredu, Alipitio [67] conducted in Ghana, Aikpitanyi, Okonofua [68] conducted in low and lower middle income countries and Kitila, Feyissa [69] conducted in Southwest Ethiopia. Every healthcare institution aims to provide excellent quality services for maternal health, resulting in increased satisfaction. Thus, if services are not of good quality, they may discourage clients from accessing them [70]. It is therefore important that the capacity of the health system in Ghana be strengthened to provide quality skilled delivery services to the people. The commitment of the government towards improving the health of

its citizens, including maternal health, will be reflected in the overall performance of the health system.

This study also shows that sociocultural and religious beliefs and practices toward birth seem to be an inherent custom in some parts of the country and hinders the effective uptake of skilled delivery. This is consistent with what is reported in the studies of [68,71]. It is therefore important that health education on the importance of skilled delivery be intensified, especially in these parts of the country. If possible, incentives in the form of monetary benefits or delivery packages should be given to women as a form of policy intervention strategies to attract women in these communities to have skilled deliveries.

It is evident from this review that some women lack autonomy in seeking skilled delivery thus, are mostly influenced by their husbands or relations to have home deliveries. In the Ghanaian culture patriarchy practice require women to seek permission before making most decisions. In the rural areas most especially where women have limited access to financial resource and education, decision making regarding seeking skilled delivery tends to be controlled by their husbands and or family members. This shows that, factors such as sociocultural and lack of women's autonomy, is largely due to poverty and illiteracy thus, must be the reason why women including their households might opt for home deliveries [53]. These sociocultural and socioeconomic factors were also reported in a cross-sectional study of Anyait, Mukanga [72] conducted in Uganda and in study of Kitila, Feyissa [69] conducted in Southwest Ethiopia. Their findings revealed that these factors have negatively impacted on utilization of skilled delivery.

To improve the effective uptake of SBA in Ghana, it is imperative that policy makers and stakeholders put in strategies to intensify health education on skilled delivery care in communities with low uptake of SBA. Efforts should also be made to remove the indirect cost associated with skilled delivery even with the 'free' delivery care policy particularly for the poorest groups so that women can freely make independent decisions relating to their choice of place of delivery. The proposed comprehensive framework in this study is to serve as a guide to policy makers and researchers working in the field of maternal health, in planning and decision making as well as researching into improving maternal health in Ghana. It also seeks to provide basic knowledge and serve as a guide for the development of future frameworks on SBA in Ghana (Fig 2).

## Strength and limitation

This study has acknowledged that the articles reviewed provided concise information relevant to the topic, however, it is possible that there are other factors other than the ones identified in this review that can also influence SBA. Hence, more research is recommended for analysis of these factors in future studies.

## Conclusion

This study explored the residual barriers to SBA in Ghana. The main factors influencing SBA explored were the health system including its related policy factors and community factors. The study showed an improvement in SBA however, barriers still exist that prevents some women from accessing skilled delivery care. One of such important barriers identified from the study is about the quality of skilled delivery. The health service delivery in the country is faced with numerous challenges, such as inadequate medical supplies and health infrastructure, inadequate and poor attitude of SBAs, inefficient health governance, which makes it difficult to improve access to health, as well as the quality of health service delivery. Notwithstanding, sociocultural practices such as lack of women's autonomy as well as

traditional belief/practices surrounding birth and socioeconomic status of women also influence SBA. This is because the free delivery care is not totally free, as indirect costs associated with facility delivery exist. This study has therefore found that, the factors influencing skilled birth utilization in Ghana are multifaceted ranging from health system, policy, community as well as individual levels, thus require a holistic approach in dealing with them if Ghana wants to achieve the SDG targets 3.1, 3.2 and 3.3.1 by 2030.

## Supporting information

**S1 Checklist. Grey and peer reviewed articles date between 2005 and 2022.**
(DOCX)

## Acknowledgments

The authors would like to acknowledge Professor Kielmann Karina, for her valuable contributions to this review.

## Author Contributions

**Conceptualization:** Juliet Abredu, Sophie Witter.

**Formal analysis:** Juliet Abredu.

**Methodology:** Juliet Abredu, Sophie Witter.

**Supervision:** Veronica Millicent Dzomeku, Sophie Witter.

**Validation:** Juliet Abredu, Catherine K. Dwumfour, Veronica Millicent Dzomeku, Sophie Witter.

**Writing – original draft:** Juliet Abredu, Catherine K. Dwumfour, Boo Alipitio, Mawusi Alordey, Veronica Millicent Dzomeku, Sophie Witter.

**Writing – review & editing:** Juliet Abredu, Catherine K. Dwumfour, Boo Alipitio, Mawusi Alordey, Veronica Millicent Dzomeku, Sophie Witter.

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
