## [Decision Letter · Decision Letter 0]

20 Nov 2023

PGPH-D-23-01209

A scoping review of the residual barriers to skilled birth attendance in Ghana: A conceptual framework and a fish bone analysis

Dear Dr. Abredu,

Thank you for submitting your manuscript to PLOS Global Public Health. After careful consideration, we feel that it has merit but does not fully meet PLOS Global Public Health’s publication criteria as it currently stands. Therefore, we invite you to submit a revised version of the manuscript that addresses the points raised during the review process.

We look forward to receiving your revised manuscript.

Kind regards,

Susan Julia Chand, PhD

Guest Editor

Journal Requirements:

1. Please amend your online Financial Disclosure statement. If you did not receive any funding for this study, please simply state: “The authors received no specific funding for this work.”

2. Please update your online Competing Interests statement. If you have no competing interests to declare, please state: “The authors have declared that no competing interests exist.”

4. Please ensure that you refer to Figure 2 in your text as, if accepted, production will need this reference to link the reader to the figure.

5. Please ensure that you refer to Table 1 in your text as, if accepted, production will need this reference to link the reader to the table.

6. We have noticed that you have uploaded Supporting Information files, but you have not included a list of legends. Please add a full list of legends for your Supporting Information files after the references list.

Additional Editor Comments (if provided):

Reviewers' comments:

Reviewer's Responses to Questions

**Comments to the Author**

1. Does this manuscript meet PLOS Global Public Health’s publication criteria? Is the manuscript technically sound, and do the data support the conclusions? The manuscript must describe methodologically and ethically rigorous research with conclusions that are appropriately drawn based on the data presented.

Reviewer #1: Yes

2. Has the statistical analysis been performed appropriately and rigorously?

Reviewer #1: I don't know

3. Have the authors made all data underlying the findings in their manuscript fully available (please refer to the Data Availability Statement at the start of the manuscript PDF file)?

Reviewer #1: Yes

4. Is the manuscript presented in an intelligible fashion and written in standard English?

Reviewer #1: Yes

5. Review Comments to the Author

Reviewer #1: Overall, this was an informative paper. I think it would be helpful for the author(s) to provide a rationale for the wide range of years from which the chosen papers were selected. It would also be helpful to make sure that the full names of all organizations that are referred to using acronyms are included.

6. PLOS authors have the option to publish the peer review history of their article (what does this mean?). If published, this will include your full peer review and any attached files.

**Do you want your identity to be public for this peer review?** For information about this choice, including consent withdrawal, please see our Privacy Policy.

Reviewer #1: No

---

## [Editor Report · Decision Letter 1]

22 Jan 2024

A scoping review of the residual barriers to skilled birth attendance in Ghana: A conceptual framework and a fish bone analysis

PGPH-D-23-01209R1

Dear Miss Abredu,

We are pleased to inform you that your manuscript 'A scoping review of the residual barriers to skilled birth attendance in Ghana: A conceptual framework and a fish bone analysis' has been provisionally accepted for publication in PLOS Global Public Health.

Best regards,

Susan Julia Chand, PhD

Guest Editor